# Error-mitigated quantum gates exceeding physical fidelities in a trapped-ion system

Shuaining Zhang [1], Yao Lu[1], Kuan Zhang[1,2], Wentao Chen[1], Ying Li [3]*, Jing-Ning Zhang[1,4]* & Kihwan Kim [1]*

Various quantum applications can be reduced to estimating expectation values, which are inevitably deviated by operational and environmental errors. Although errors can be tackled by quantum error correction, the overheads are far from being affordable for near-term technologies. To alleviate the detrimental effects of errors on the estimation of expectation values, quantum error mitigation techniques have been proposed, which require no additional qubit resources. Here we benchmark the performance of a quantum error mitigation technique based on probabilistic error cancellation in a trapped-ion system. Our results clearly show that effective gate fidelities exceed physical fidelities, i.e., we surpass the break-even point of eliminating gate errors, by programming quantum circuits. The error rates are effectively reduced from $(1.10 \pm 0.12) \times 10^{-3}$ to $(1.44 \pm 5.28) \times 10^{-5}$ and from $(0.99 \pm 0.06) \times 10^{-2}$ to $(0.96 \pm 0.10) \times 10^{-3}$ for single- and two-qubit gates, respectively. Our demonstration opens up the possibility of implementing high-fidelity computations on a near-term noisy quantum device.

[1] Center for Quantum Information, Institute for Interdisciplinary Information Sciences, Tsinghua University, Beijing 100084, China. [2] MOE Key Laboratory of Fundamental Physical Quantities Measurements, Hubei Key Laboratory of Gravitation and Quantum Physics, PGMF and School of Physics, Huazhong University of Science and Technology, Wuhan 430074, China. [3] Graduate School of China Academy of Engineering Physics, Beijing 100193, China. [4] Beijing Academy of Quantum Information Sciences, Beijing 100193, China. *email: yli@gscaep.ac.cn; zhangjn@baqis.ac.cn; kimkihwan@mail.tsinghua.edu.cn

 **1**

Quantum computers[1] can extend classical computational reach in diverse research fields, including quantum chemistry, material science, and even machine learning. Based on various technological advances so far, such nontrival quantum applications have been pursued with currently available devices mainly through quantum-classical hybrid schemes[2,3]. The schemes combine the advantages of classical and quantum computation, where quantum processors are used to estimate expectation values of physical observables on certain states for classical feedback. The hybrid schemes can be applied to estimate the ground-state energies of molecules[3–5], to simulate quantum models in materials[6] and high-energy physics[7], and to find approximate solutions of optimization problems[8]. Although it is anticipated that around a hundred well-behaved qubits are required for such schemes to outperform current classical counterparts in quantum chemistry[9–11], the advantages are only possible with accurate quantum processors. However, expectation values obtained with output results of the quantum devices are inevitably deviated because of errors originated from both environmental fluctuations and operational imperfections. Therefore, techniques for accurately estimating expectation values with improving the accuracy of noisy quantum processors are of great importance.

Apart from physically improving the devices, the deviations in estimating expectation values can be suppressed on the algorithmic level. For example, quantum error correction[12,13] provides a mean of fault-tolerant quantum computation, which results in accurate expectation values. However, quantum error correcting codes require complex coding schemes, a large number of physical qubits, and low error rates, which are still far from being affordable for near-term quantum technologies[14,15]. Consequently, it has not yet been demonstrated that quantum fault tolerance protocols can increase the fidelity of computation operations in any physical implementation. Alternatively, for the quantum algorithms estimating expectation values, the reliability of computation result can be improved by recently proposed error mitigation schemes[16–20] without challenging requirements for quantum error corrections. The probabilistic error-cancellation method provides a comprehensive way to mitigate errors in expectation estimation tasks[17,18,21]. It begins with characterizing imperfect operations on the quantum device by tomography technique and then cancels errors by sampling random quantum circuits, according to a quasi-probability distribution derived from reconstructing ideal quantum operations with characterized imperfect ones. Please note that this method does not improve the physical quality of quantum states or gates but reduces the error in the estimation of expectation values.

Here we construct a trapped-ion system with full controllability and investigate the universal validity of the probabilistic error-cancellation method in a general quantum computational context. We apply the method to every imperfect elementary quantum operation and benchmark the performance of error-mitigated quantum computation[22]. We observe singnificant improvements on effective gate fidelities of single- and two-qubit gates by an order of magnitude to those of physical gates. Here, the effective gate fidelities are obtained by fitting the corresponding expectation values estimated with error mitigation, which are not actual physical gate fidelities.

## Results

### Paradigm of error-mitigated quantum computation.
The paradigm of error-mitigated quantum computation is shown in Fig. 1. The noisy quantum device is treated as a multi-qubit black box in Fig. 1a, capable of preparing each qubit into an initial state $\rho_0$, performing a set of single-qubit and two-qubit gates, and two-

outcome measurement on each qubit, which is described by a positive operator-valued measure $\mathcal{M} \equiv \{E_0, I - E_0\}$ with $I$ being the $2 \times 2$ identity operator. These quantum operations are generally not accurate because of errors from operational imperfections and environmental fluctuations. As proposed in ref. [18], we perform the gate set tomography[23–25] and characterize state preparation and measurement (SPAM) and gates of noisy quantum devices by Gram matrices and Pauli transfer matrices (PTMs), respectively[25], as shown in Fig. 1b. When we repeatedly execute a quantum circuit with such a noisy device aiming at obtaining the expectation values of observables of interest, the estimation will be deviated from the ideal case due to the imperfection of the quantum device, as shown in Fig. 1c. The correction of each noisy quantum operation can be decomposed to the combination of experimental basis operations (which we give later) with quasi-probabilities as shown in Fig. 1d. As some of the quasi-probabilities can be negative, we cannot physically implement the decomposition. However, the expectation of the decomposition can be estimated by sampling circuits with random basis operations according to the quasi-probabilities[17,18]. After running the random circuits with the corrections, the probability distribution of the output expectation value is shifted towards the ideal value at a cost of enlarged variance due to the presence of negative values in the quasi-probabilities[18], as shown in Fig. 1c. The variance can be reduced by increasing the repetition number, which is the number of random-circuit instances.

### Experimental realization.
In our experimental realization, the quantum hardware encapsulated in the black box is a trapped-ion system, where $^{171}\mathrm{Yb}^+$ ions are trapped into a linear crystal and individually manipulated by global and individual laser beams, as shown in Fig. 1a. To encode quantum information, a pair of clock states in the ground-state manifold $^2S_{1/2}$, i.e., $|F = 0, m_F = 0\rangle$ and $|F = 1, m_F = 0\rangle$, are denoted as the computational basis $\{|0\rangle, |1\rangle\}$ of a qubit. At the beginning of executing a quantum circuit, each ion qubit is initialized to $|0\rangle$ by optical pumping. We implement single-qubit operations by Raman laser beams with beatnote frequency about the hyperfine splitting $\omega_0 = 2\pi \times 12.642821$ GHz. In addition, the two-qubit operation, i.e., the Mølmer-Sørensen $YY$-gate ($\mathrm{MS}_{YY}$) is realized by driving transverse motional modes[26,27], with frequencies in the $x$-direction $\{v_1, v_2\} = \{1.954, 2.048\}$ MHz. We apply amplitude-shaped[28] bichromatic Raman beams with beatnote frequencies $\omega_0 \pm \mu$, where $\mu$ is set to be the middle frequency of the two motional modes, and achieve the $\mathrm{MS}_{YY}$ gate for 25 μs. We also realize the MS $ZZ$-gate ($\mathrm{MS}_{ZZ}$) by adding single-qubit rotations before and after the $\mathrm{MS}_{YY}$ gate[29] (see Supplementary Fig. 4b). At the end of the execution, internal states of qubits are measured by state-dependent fluorescence detection[30]. It is noteworthy that to collect fluorescence photons, we use a photomultiplier tube in the single-qubit case and an electron-multiplying charge-coupled device (EMCCD) in the two-qubit case.

### Characterization of quantum device.
We introduce the PTM representation for the mathematical description of an $n$-qubit noisy quantum device, where density operators $\rho$ and physical observable $E$ are represented by $2^n$-entry column vectors $|\rho\rangle\rangle$ and row vectors $\langle\langle E|$, and quantum gates $G$ are represented by $2^{2n} \times 2^{2n}$ PTMs $R_G$. Here, the expectation value of the observable $\hat{E}$ after operating $G_s$ on the initial state $\hat{\rho}$ is represented by $\langle\langle E|R_G|\rho\rangle\rangle$. PTMs can be determined by gate set tomography, which requires informationally complete data obtained from experiments with initial states from a basis set $\mathcal{S}_n \equiv \{|0\rangle, |1\rangle, |1\rangle_X, |1\rangle_Y\}^{\otimes n}$ and measurement of the observables from

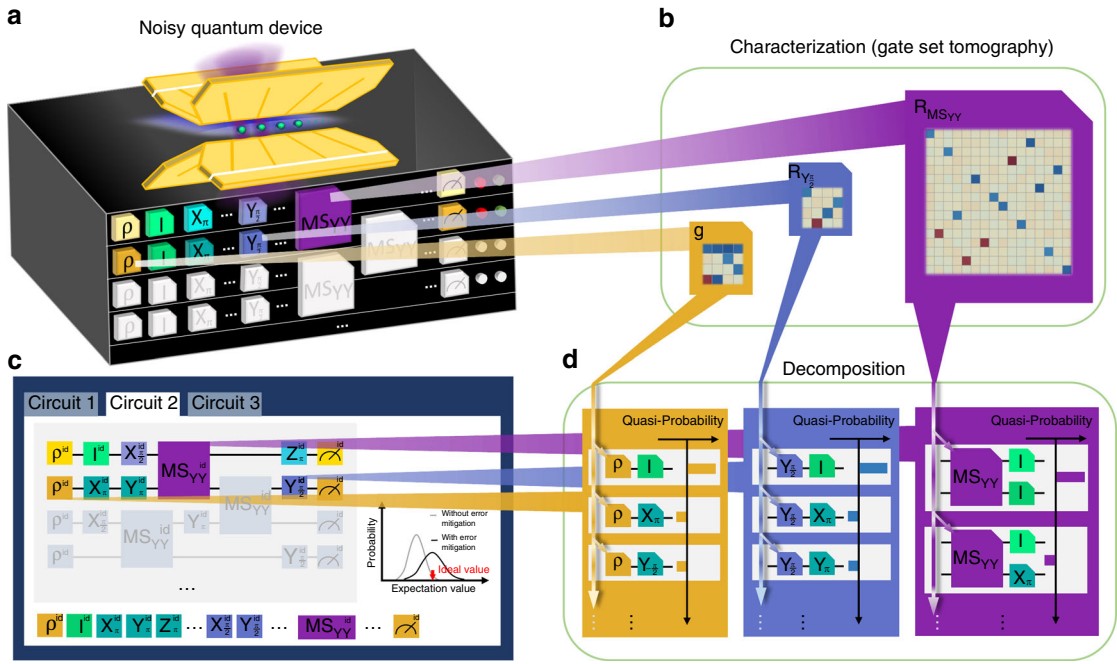

**Fig. 1 Paradigm of error-mitigated quantum computation. a** Quantum black box based on a trapped $^{171}$Yb$^+$-ion system. Each button on the surface corresponds to an operation exerted on the quantum system encapsulated, where the buttons with $\rho$ and $M$ are for initial state preparation and computational basis measurement, whose results are indicated by the lights. The other buttons are for single-qubit and two-qubit quantum operations on certain qubits. The operations are implemented by global (blue) and individual (purple) laser beams illuminating the ion qubits. **b** Characterization of the quantum black box. The error-affected state preparation and measurement is characterized by the Gram matrix $g$ and the effect of each operation $G$, such as $Y_{\frac{\pi}{2}}$ and MS$_{YY}$, is described by a Pauli transfer matrix $R_G$ in the superoperator formalism, which is obtained by gate set tomography. **c** Construction of unbiased estimator of an expectation value specified by a quantum circuit, with building blocks including initial state preparation, different single-qubit and two-qubit gates, and the final measurement. With error mitigation, the distribution of the output expectation value is shifted towards the ideal value at a cost of enlarged variance. **d** Quasi-probability decomposition for the ideal initial state and exemplary single-qubit and two-qubit gates. As the errors in state preparation and those in measurement are indistinguishable, we ascribe both of the errors to state preparation and decompose the ideal initial state with a set of experimental basis states, prepared by state initialization followed by a random fiducial gate. The PTM of an ideal quantum gate can be expanded as a quasi-probability distribution over random gate sequences consisting of the experimental gate and one of the experimental basis operations, Pauli operations in the experiment. The error-mitigated estimation of the expectation value is then obtained by the Monte-Carlo method (see Methods).

the $n$-qubit Pauli basis $\mathcal{P}_n = \{I, X, Y, Z\}^{\otimes n}$, where $|1\rangle_X$ and $|1\rangle_Y$ are the eigenstates of Puali operators $X$ and $Y$, respectively. Compared with quantum process tomography, gate set tomography is featured by appropriately taking consideration of SPAM errors, which is of great importance in quantum computations with high accuracy. In gate set tomography, the states in $\mathcal{S}_n$ and the measurement of observables in $\mathcal{P}_n$ are realized by using a set of fiducial gates $\mathcal{F}_n \equiv \{I, X_\pi, Y_{-\frac{\pi}{2}}, X_{\frac{\pi}{2}}\}^{\otimes n}$ consisting of the identity operation and the $X$ or $Y$ axis rotations on each qubit, which are to be characterized together with the rest of the quantum operations. The single-qubit SPAM errors are reflected in the Gram matrix[25], as shown in Fig. 2a, which is obtained by preparing the qubit in one of the states $\mathcal{S}_1$, $|\rho_i\rangle\rangle = R_{F_i}|\rho_0\rangle\rangle$, and measuring the expectation values of the operators in the single-qubit Pauli basis $\mathcal{P}_1$, $\langle\langle E_i| = \langle\langle E_0|R_{F_i}$, where $\rho_0$ and $E_0$ are ideally associated with $|0\rangle\langle0|$ and $Z$, respectively.

For single-qubit randomized benchmarking[22], we design pulse sequences for implementing major-axis $\pi$ pulses $\{X_{\pm\pi}, Y_{\pm\pi}, Z_{\pm\pi}\}$ and $\frac{\pi}{2}$ pulses $\{X_{\pm\frac{\pi}{2}}, Y_{\pm\frac{\pi}{2}}\}$. Thus, the gate set for the single-qubit case is $\mathcal{G}_1 = \{I, X_{\pm\pi}, Y_{\pm\pi}, Z_{\pm\pi}, X_{\pm\frac{\pi}{2}}, Y_{\pm\frac{\pi}{2}}\}$, where $I$ is the identity operation. The gate set for implementing two-qubit random circuits are $\mathcal{G}_2 = \mathcal{G}_1^{\otimes 2} \cup \{MS_{YY}, MS_{ZZ}\}$. We experimentally obtain the PTMs of all the gates in the gate set by maximizing a likelihood function with the assumption that Pauli errors are dominant in our devices (see Methods).

The reconstructed PTMs of $X_{\pm\frac{\pi}{2}}$ and $Y_{\pm\frac{\pi}{2}}$ for the single-qubit case and those of MS$_{YY}$ and MS$_{ZZ}$ gates for the two-qubit case are shown in Fig. 2b, c, respectively (more data for the single-qubit case are in Supplementary Fig. 1a). We note that, for the gate set tomography of two qubits, we apply a two-step parameter estimation, as the infidelities for the single-qubit gates are about an order lower than those of the two-qubit gates. We first determine the Pauli error rates for all the single-qubit gates in $\mathcal{G}_1^{\otimes 2}$ as described above and then characterize the two-qubit gate MS$_{YY}$ based on the knowledge of the characterized single-qubit gates (see Methods). The MS$_{ZZ}$ gate is derived from those results. Using these reconstructed PTMs, we numerically simulate the single-qubit randomized benchmarking and two-qubit random circuits on a classical computer. The comparisons between the numerically reconstructed and experimental data clearly validate the Pauli error assumption within both error bars (see Supplementary Fig. 2).

The initial state, quantum gates, and measurement are deviated from the ideal ones, as experimentally characterized by Gram matrix and PTMs. Mathematically, we can reconstruct the ideal ones by a weighted combination of experimental operations[17,18]. As we cannot distinguish errors in state preparation from those in measurement, we ascribe all of the SPAM errors to state preparation and decompose the initial state $|\rho_0^{id}\rangle\rangle = \sum_i q_{0,i}|\rho_i\rangle\rangle$. The quasi-probabilities $q_{0,i}$ for the decomposition of the ideal single-qubit initial state is shown in Fig. 3a. It is noteworthy that

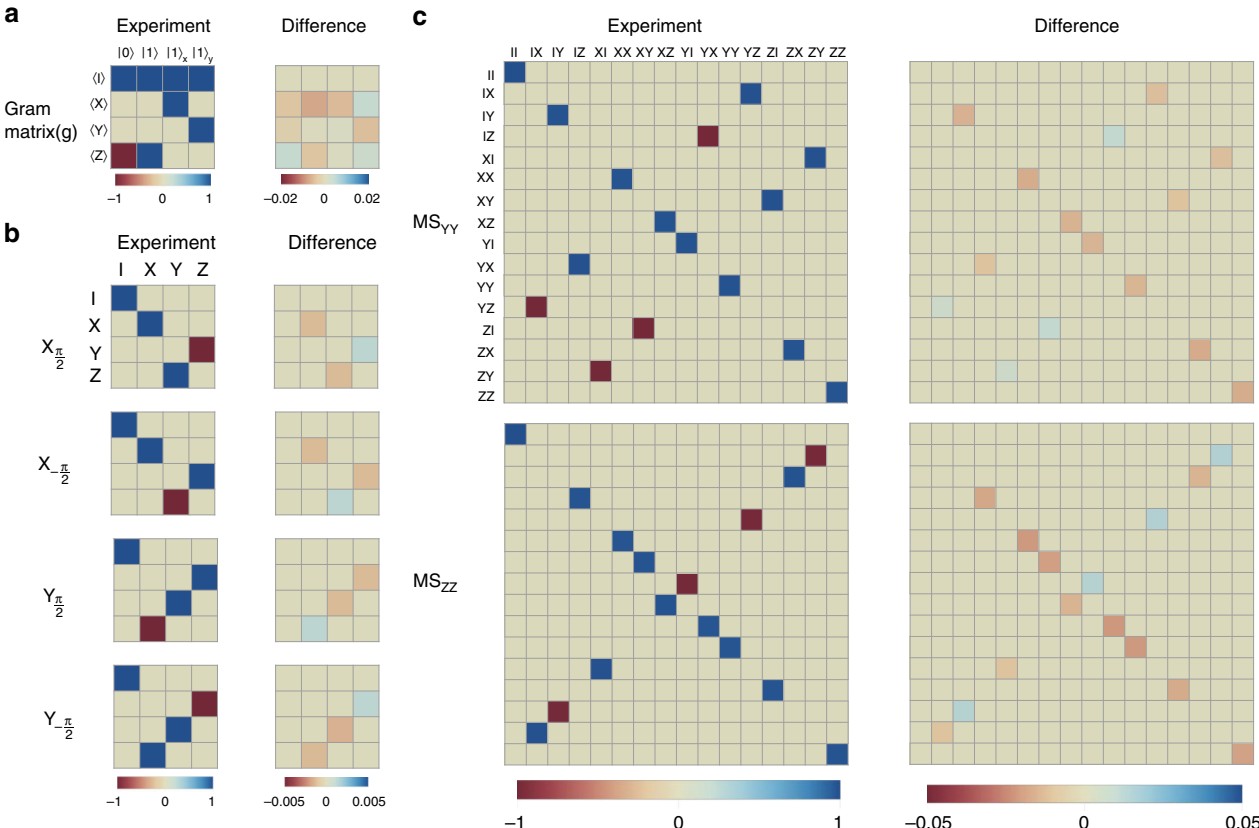

**Fig. 2 Characterization of noisy quantum devices obtained by gate set tomography. a** Gram matrix and **b** PTMs of single-qubit gates for the single-qubit case. The single-qubit experiments are implemented with a single trapped ion. The Gram matrix characterize the SPAM error, which is obtained by preparing states in $\mathcal{S}_1$ and measuring expectation values of operators in $\mathcal{P}_1$. Here we show the PTMs of experimental gates $X_{\pm\frac{\pi}{2}}$ and $Y_{\pm\frac{\pi}{2}}$, the so-called computational gates in randomized benchmarking, as examples (PTMs of other experimental gates are shown in Supplementary Fig. 1). **c** Pauli transfer matrices of the experimental gates $MS_{YY}$ and $MS_{ZZ}$ in the two-qubit case. It is worth noting that we calibrate the SPAM errors as proposed in ref. [31] and the PTMs of single-qubit gates on both qubits (not shown) are not noticeably different to those for the single-qubit case. In each subfigure, the left column shows the experimentally obtained matrices and the right column shows the difference between the experimental and the ideal matrices, i.e., $R_G - R_G^{id}$ with $G$ being one of the quantum operations being characterized.

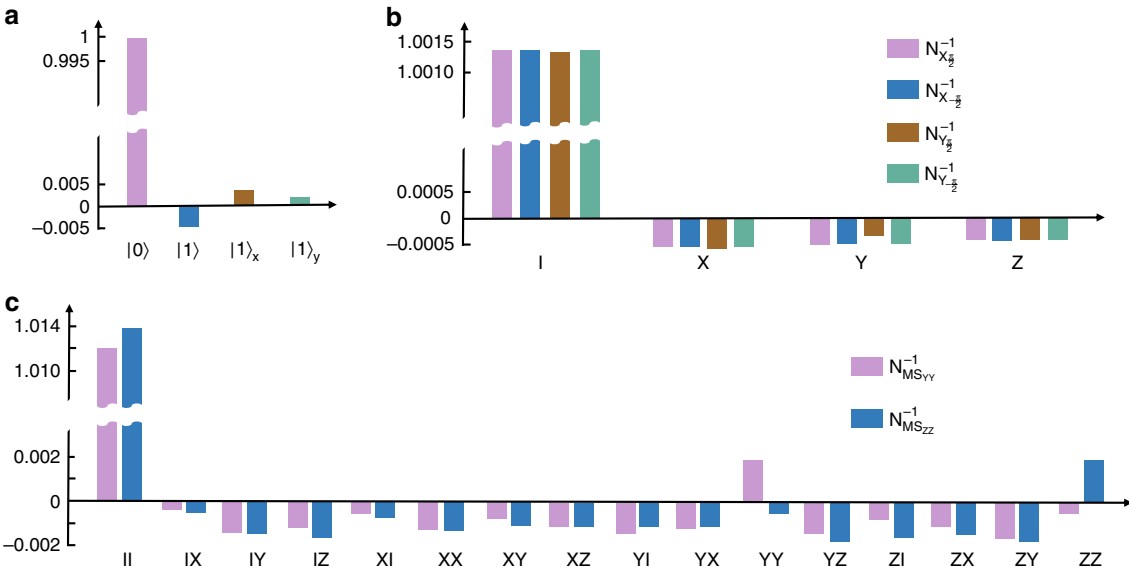

**Fig. 3 Quasi-probability decomposition. a** Quasi-probabilities in the decomposition of the ideal single-qubit initial state with experimental initial states in $\mathcal{S}_1$. **b** Quasi-probabilities in the decomposition of the inverse noise operations of the four experimental single-qubit gates $\{X_{\pm\frac{\pi}{2}}, Y_{\pm\frac{\pi}{2}}\}$. **c** The same as **b** for the experimental two-qubit gates $\{MS_{YY}, MS_{ZZ}\}$.

for the two-qubit case, the SPAM errors are much more serious because of the EMCCD and we calibrate the results to remove the SPAM errors as proposed in ref. [31]. We prepare the system in the states $|00\rangle$ and $|11\rangle$, and measure the state fidelities of $|0\rangle$ and $|1\rangle$ for both qubits. The infidelities of these states give the SPAM error probability associated with each measurement outcome, which can then be used to remove the SPAM errors by data processing.

An ideal quantum gate $G_s^{id}$ can be written as the experimental one followed by the inverse of noise operation, i.e., $R_{G_s^{id}} = N_s^{-1} R_{G_s}$, where the noise operation $N_s$ introduces errors in the experimental gate $R_{G_s} = N_s R_{G_s}^{id}$. The inverse of the noise operation $N_s^{-1}$ is then decomposed by the experimental operations associated with the $n$-qubit Pauli group, $N_s^{-1} = \sum_j q_{s,j} R_{P_j}$ with Pauli error assumption, where the quasi-probabilities $q_{s,j}$ are determined by a set of linear equations. We show decompositions of the inverse error operations for single-qubit gates $\{X_{\pm\frac{\pi}{2}}, Y_{\pm\frac{\pi}{2}}\}$ in Fig. 3b (more data in Supplementary Fig. 1b) and for two-qubit gates $\{MS_{YY}$ and $MS_{ZZ}\}$ in Fig. 3c.

**Benchmarking of the quantum error mitigation protocol**. We benchmark the performance of the quantum error mitigation using a set of random computations, in the spirit of randomized benchmarking. Each specific computation starts with fully polarized initial states, $|0\rangle$ in the single-qubit case and $|00\rangle$ in the two-qubit case, and ends with measuring $Z$ on each qubit. Between the SPAM, there is a sequence of randomly selected quantum gates. We note that the randomness in selecting the gate sequence is for the purpose of benchmarking the performance rather than correcting errors. For each specific computation, i.e., gate sequence, we apply the error mitigation and modify the gate sequence with random basis operations to correct errors. We remark that, for each specific computation, we observe the improvement on the computation accuracy by using the error mitigation.

For the single-qubit case, benchmarking computations are selected according to the standard randomized benchmarking, i.e., a gate sequence of length $L$ contains $L$ computational gates and $L+1$ interleaving identity or Pauli operations, uniformly drawn from the set $\{X_{\pm\frac{\pi}{2}}, Y_{\pm\frac{\pi}{2}}\}$ and $\{I, X_{\pm\pi}, Y_{\pm\pi}, Z_{\pm\pi}\}$, respectively. For each sequence length $L$, we choose four sequences whose ideal final states are the eigenstates of the Pauli $Z$ operator. We then

repeatedly implement each of the sequences with a trapped-ion system consisting of a single trapped ion and measure the state fidelity between the ideal and experimentally prepared final states. In Fig. 4a, we show the dependence of the average fidelity without error mitigation, obtained by averaging the state fidelities over sequeces of the same length, on the sequence length. We numerically fit the average fidelity with an exponential function and obtain the error rate per single-qubit gate as $(1.10 \pm 0.12) \times 10^{-3}$.

To obtain unbiased estimator of the expectation value, both the initial state and $2L+1$ gates in the selected sequence need to be decomposed and resampled, where the initial state is replaced probabilistically by one of the states in $\mathcal{S}_1$, and each experimental gate is followed by a random Pauli or identity operation drawn from $\mathcal{P}_1$. Thus, for a specific computation with $(2L+1)$ gates, there are $4^{2L+2}$ possible experimental settings. As the number of settings grows exponentially with the length of the random sequence, we use the Monte-Carlo method to compute the result by sampling random experimental settings, which are specified by an index $i$ for the initial state $|\rho_i\rangle$ and two $(2L+1)$-entry index vectors $\mathbf{a} \equiv (a_1, \dots, a_{2L+1})$ and $\mathbf{b} \equiv (b_1, \dots, b_{2L+1})$ specifying the computation and the choices of the error-compensating operations. We note that for a specific computation, $\mathbf{a}$ is determined but $\mathbf{b}$ is random. The probability of an experimental setting $\langle\langle E_0| \prod_{l=1}^{2L+1} R_{P_{b_l}} R_{G_{a_l}} |\rho_i\rangle\rangle$, where $G_{a_l} \in \mathcal{G}_1$ and $P_{b_l} \in \mathcal{P}_1$, is $C^{-1}\left|q_{0,i}\left(\prod_{l=1}^{2L+1} q_{a_l,b_l}\right)\right|$. Here, the rescaling factor $C = \sum_{i,\dots,(a_l,b_l),\dots}\left|q_{0,i}\left(\prod_{l=1}^{2L+1} q_{a_l,b_l}\right)\right| \geq 1$ characterizes the cost to mitigate the errors. It is noteworthy that the signs of the coefficients, i.e., $\text{sgn}\left[q_{0,i}\left(\prod_{l=1}^{2L+1} q_{a_l,b_l}\right)\right]$, are integrated into the measurement results of the random experiments (see Methods). In Fig. 4a, we represent the error-mitigated single-qubit randomized benchmarking with length $L$ up to 64 and show that the single-qubit gate error rate is effectively suppressed to $(1.44 \pm 5.28) \times 10^{-5}$.

For the two-qubit case, we select four gate sequences as benchmarking computations for each length $L$. Each sequence contains $L$ two-qubit gates uniformly drawn from the set $\{MS_{YY}, MS_{ZZ}\}$ with interleaving single-qubit gates[32]. The sequence is selected under the restriction that the ideal final state is an eigenstate of $Z^{\otimes 2}$. Similar to the single-qubit case

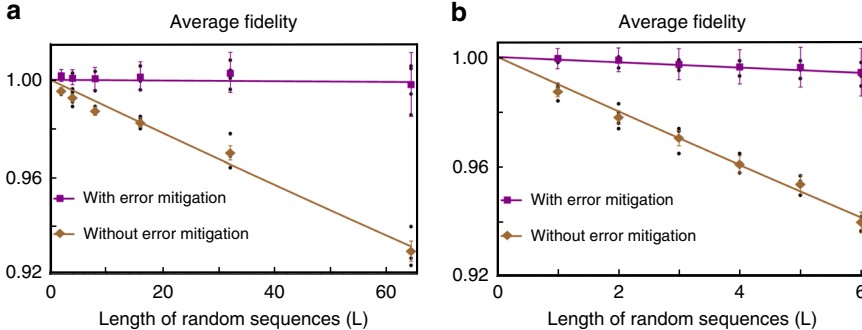

**Fig. 4 Experimental data for error-mitigated quantum computation. a** The single-qubit randomized benchmarking. The data points with (purple square) and without (yellow diamond) error mitigation are obtained from averaging the final-state fidelitiy over different random sequences of the same length (black dots). The error bars are the SD of the average fidelities computed using the formula of uncertainty propagation. The solid lines, obtained by fitting, show the exponential decrease of the average fidelities, indicating the physical and effective average errors per gate being $(1.10 \pm 0.12) \times 10^{-3}$ and $(1.44 \pm 5.28) \times 10^{-5}$, respectively. Please note that some of the fidelities with error mitigation are larger than 1 because of the rescaling factor $C > 1$ (see main text and Methods) and the limited sampling for data points. Although the current protocol does not guarantee a physical outcome, the error mitigation procedure shifts the distribution of the computation result towards the true value with a large enough sampling. **b** The two-qubit random-circuit computation. Decay rates indicated by the average fidelity curves without and with error mitigation are $(0.99 \pm 0.06) \times 10^{-2}$ and $(0.96 \pm 0.10) \times 10^{-3}$, respectively.

described above, we apply error mitigation to each of the two-qubit gate sequences with length $L$ up to 6 and represent the error-mitigated results in Fig. 4b, where the two-qubit gate error rate is effectively suppressed from $(0.99 \pm 0.06) \times 10^{-2}$ to $(0.96 \pm 0.10) \times 10^{-3}$.

## Discussion

Our work shows that for the estimation of expection values, the error mitigation technique, i.e., probabilistic error cancellation[17,18,21], surely have the capacity of surpassing the break-even point, where the effective gates are superior to their physical building blocks, at an affordable cost with respect to near-future quantum techniques. We note that error mitigation techniques are developed for the intermediate-scale quantum computation. The cost of the error mitigation increases with the circuit depth; therefore, techniques such as quantum error correction are still needed for large-scale fault-tolerant quantum computation. The effective infidelity after error mitigation comes from the Pauli error assumption, time-dependent systematic drifting[33] for both single-qubit and two-qubit cases, and crosstalk error of single-qubit addressing operations for the two-qubit case (see Methods). Thus, further improvement requires both calibrating and stabilizing the quantum device. With technologies to tackle the crosstalk error, the probabilistic error-cancellation method of quantum error mitigation can be straightforwardly applied to systems with more qubits for realizing high-fidelity quantum computation.

## Methods

**Maximum-likelihood gate set tomography**. To obtain the PTMs of all the gates in the gate set, we experimently measure informationally complete data consisting of the average $\bar{m}_{ijk}$ and variance $\Delta_{ijk}$ of the expectation value $\langle\langle E_i | R_{G_j} | \rho_k \rangle\rangle$, which are obtained by repeating the corresponding experimental settings enough number of times. We assume Pauli errors are dominant in our device, where each of the noisy quantum gate $G_j \in \mathcal{G}_n$ is modeled with the ideal gate $G_j^{id}$ followed by a Pauli error channel. We use a maximum-likelihood estimation for the reconstruction of PTMs of all the gates in the gate set, parameterized as ansatz $R_{G_j} = N_j R_{G_j}^{id}$, where $N_j = \sum_l p_{j,l} R_{P_l}$, with variational parameters being gate-specific Pauli error rates $p_{j,l}$. With the ansatz for each gate, we calculate the ansatz prediction for the expectation value of each experimental setting, denoted as $m_{ijk}$. We then define the following likelihood function[25],

$$\mathcal{L} = \prod_{i,j,k} \exp\left[-(m_{ijk} - \bar{m}_{ijk})^2 / \Delta_{ijk}^2\right], \quad (1)$$

which takes its maximum value when the experimental average values $\bar{m}_{ijk}$ and the ansatz expectations $m_{ijk}$ coincide with each other. Thus, the gate-specific Pauli error rates can be determined by maximizing the likelihood function, with which we construct the PTMs of the imperfect gates that are implementable in the quantum device.

**Characterization and decomposition of single-qubit gate set**. We use gate set tomography to characterize the single-qubit operations. In the superoperator formalism, each experimental single-qubit operation $R_{G_s}$ can be describe as an ideal $4 \times 4$ PTM followed by a PTM of noise operation $N_s$. With Pauli error assumption, each $N_s$ can be written as $N_s = p_{s,0} R_I^{id} + p_{s,1} R_X^{id} + p_{s,2} R_Y^{id} + p_{s,3} R_Z^{id}$, where $p_{s,j}$ are the Pauli error rates and $\sum_j p_{s,j} = 1$ for trace-preserving condition. As there are 11 gate in $\mathcal{G}_1$, $\mathcal{F}_1 \subset \mathcal{G}_1$ and the experimental initial state $\rho_0$ can be characterized by 3 parameters, we need to obtain the values for $11 \times 3 + 3 = 36$ parameters. We run $3 \times 11 \times 4$ different experimental settings specified by $\langle\langle E_0 | R_{F_k} R_{G_j} R_{F_i} | \rho_0 \rangle\rangle$ with repetitions of 10,000 per setting to collect experimental data $\bar{m}_{ijk}$, where $i = 1, ..., 4$ for state preparation, $j = 1, ..., 11$, and $k = 1, 2, 3$ for different measurement settings. The ansatz prediction $m_{ijk} = \langle\langle E_0 | N_{F_k} R_{F_k}^{id} N_{G_j} R_{G_j}^{id} N_{F_i} R_{F_i}^{id} | \rho_0 \rangle\rangle$ contain Pauli error rates as variational parameters, which we numerically optimize to maximize the likelihood function in Eq. (1). The obtained PTMs are shown in Fig. 2b and Supplementary Fig. 1a.

Once we get experimental PTMs for single-qubit operations, we can derive the inverse of PTM of the noise operation as $N_s^{-1} = R_{G_s}^{id} R_{G_s}^{-1}$, which can be decomposed by the combination of PTMs of experimental Pauli operations with $N_s^{-1} = q_{s,0} R_I + q_{s,1} R_X + q_{s,2} R_Y + q_{s,3} R_Z$. Then, the ideal operation can be

decomposed by experimental operations as
$$R_{G_s}^{id} = q_{s,0} R_I R_{G_s} + q_{s,1} R_X R_{G_s} + q_{s,2} R_Y R_{G_s} + q_{s,3} R_Z R_{G_s}.$$

**Characterization of the two-qubit gate set**. The two-qubit gate set, i.e., $\mathcal{G}_2$ includes single-qubit operations in $\mathcal{G}_1^{\otimes 2}$ and two-qubit operations $\{MS_{YY}$ and $MS_{ZZ}\}$. As infidelities for the single-qubit gates are about an order lower than those of the two-qubit gates, it is reasonable to divide the maximum-likelihood estimation into two steps.

First, we treat each qubit in the two-qubit system as a single-qubit system and characterize the single-qubit gate set $\mathcal{G}_1$ by gate set tomography, obtaining single-qubit PTMs. The two-qubit PTMs of the single-qubit operations in $\mathcal{G}_1^{\otimes 2}$ is then obtained by a direct product of the single-qubit PTMs on both qubits. As the fiducial set $\mathcal{F}_2 \in \mathcal{G}_1^{\otimes 2}$, the PTMs of the fiducial operations are determined at this step.

Second, we characterize the native two-qubit $MS_{YY}$ gate. Under the Pauli error assumption, the PTM of the experimental $MS_{YY}$ gate is decomposed as $R_{MS_{YY}} = N_{MS_{YY}} R_{MS_{YY}}^{id}$, where $N_{MS_{YY}}$ is the PTM of the Pauli error channel containing 16 two-qubit Pauli components. After considering the trace-preserving constraint, $N_{MS_{YY}}$ has 15 parameters, which are determined by linear equations connecting the ansatz predition $\langle\langle E_0^{\otimes 2} | R_{F_k} N_{MS_{YY}} R_{MS_{YY}}^{id} R_{F_i} | \rho_0^{(1)} \otimes \rho_0^{(2)} \rangle\rangle$ and corresponding experimental results. To minimize the projection error, we choose 15 linearly independent equations out of $16 \times 9$ different settings, with most of the measured probabilities close to 0 or 1. Supplementary Fig. 3 shows the corresponding circuits for the experimental settings.

As the $MS_{ZZ}$ is implemented by a $MS_{YY}$ gate sandwiched by proper single-qubit gates, the PTM of the experimental $MS_{ZZ}$ gate is obtained by multiplying the PTMs of the corresponding experimental operations, i.e., $R_{MS_{ZZ}} = R_{X_{-\frac{\pi}{2}} \otimes X_{-\frac{\pi}{2}}} R_{MS_{YY}} R_{X_{\frac{\pi}{2}} \otimes X_{\frac{\pi}{2}}}$.

**Probabilistic error-cancellation scheme**. The concrete procedure of applying the probabilistic error cancellation to a given quantum computation task consists of the so-called characterization and calculation phases. The characterization phase is described above. In the calculation phase, we estimate expectation values of quantum circuits with the characterized imperfect quantum device. We first write down the unbiased estimator of the expectation value of a specific quantum circuit as $\langle\langle E_0^{id} | R_{G_{a_L}}^{id} \cdots R_{G_{a_1}}^{id} | \rho_0^{id} \rangle\rangle$, which can be expanded with the quasi-probability distributions obtained in the characterization phase as follows,

$$\langle\langle E_0^{id} | R_{G_{a_L}}^{id} \cdots R_{G_{a_1}}^{id} | \rho_0^{id} \rangle\rangle = \sum_i \sum_{b_1, ..., b_L} q_{0,i} q_{a_1,b_1} \cdots q_{a_L,b_L} \langle\langle E_0^{id} | R_{P_{b_L}} R_{G_{a_L}} \cdots R_{P_{b_1}} R_{G_{a_1}} | \rho_i \rangle\rangle, \quad (2)$$

where the expection value of $\langle\langle E_0^{id} | R_{P_{b_l}} R_{G_{a_L}} \cdots R_{P_{b_1}} R_{G_{a_1}} | \rho_i \rangle\rangle$ can be obtained by repeating the specific experimental setting and averaging the measurement results. The straightforward way to evaluate the unbiased estimator is summing over all possible settings. However, this is impractical, because the number of settings grows exponentially with the circuit depth. To alleviate the exponential growth, we rewrite the above expansion as a probability distribution as follows,

$$\langle\langle E_0^{id} | R_{G_{a_L}}^{id} \cdots R_{G_{a_1}}^{id} | \rho_0^{id} \rangle\rangle = C_\mathbf{a} \sum_{i,\mathbf{b}} P_\mathbf{a}(i, \mathbf{b}) g(i, \mathbf{a}, \mathbf{b}) \langle\langle E_0^{id} | R_{P_{b_L}} R_{G_{a_L}} \cdots R_{P_{b_1}} R_{G_{a_1}} | \rho_i \rangle\rangle, \quad (3)$$

with the short-hand notations $\mathbf{a} \equiv (a_1, ..., a_L)$ and $\mathbf{b} \equiv (b_1, ..., b_L)$, where $C_\mathbf{a} \equiv \sum_{i,\mathbf{b}} |q_{0,i}| \prod_l |q_{a_l,b_l}|$ is the rescaling factor, $P_\mathbf{a}(i, \mathbf{b}) = |q_{0,i}| \prod_l |q_{a_l,b_l}| / C$ is the probability distribution, and $g(i, \mathbf{a}, \mathbf{b}) = \text{sgn}(q_{0,i} \prod_l q_{a_l,b_l})$ is the sign of the setting. Then, we use important sampling to generate $M$ experimental settings, specified by $(i_m, \mathbf{b}_m)$ with $m = 1, ..., M$, according to the probability distribution $P_\mathbf{a}(i, \mathbf{b})$, and calculate the expectation value as follows,

$$\langle\langle E_0^{id} | R_{G_{a_L}}^{id} \cdots R_{G_{a_1}}^{id} | \rho_0^{id} \rangle\rangle = \frac{C_\mathbf{a}}{M} \sum_{m=1}^M g(i_m, \mathbf{a}, \mathbf{b}_m) O(i_m, \mathbf{a}, \mathbf{b}_m), \quad (4)$$

where $O(i_m, \mathbf{a}, \mathbf{b}_m)$ is the result of the projective measurement of the $m$-th setting, being either 0 or 1 in our experiment.

**Simple example**. In this section, we provide an illustrative example of applying the probabilistic error-cancellation technique to a simple quantum circuit. Suppose an experimenter plans to apply an ideal gate $G^{id} \equiv [e^{-iY\frac{\pi}{4}}]$ on an ideal initial state $\rho^{id} \equiv |0\rangle\langle 0|$ and get the ideal expectation value of observable $\langle X \rangle^{id} \equiv Tr[XG(\rho^{id})] = 1$. However, as an example of a noisy quantum device, the actual initial state could be $\rho = 90\%|0\rangle\langle 0| + 10\%\frac{I}{2}$ and the actual gate could be $G = 80\%G^{id} + 20\%D$, where $D(\rho) = \frac{I}{2}$. Then, the actual result is $\langle X \rangle = Tr[XG(\rho)] = 72\%$. With the error-cancellation procedure, the ideal initial state is decomposed as $\rho^{id} = (\rho - 10\%\frac{I}{2})/90\%$ and the ideal gate is decomposed as $G^{id} = (G - 20\%D)/80\%$. Then, the ideal expectation value can be obtained by $\langle X \rangle^{id} = Tr[XG(\rho)] \times (1/72\%) - Tr[XG(\frac{I}{2})] \times (10\%/72\%) - Tr[XD(\rho)] \times (20\%/72\%) - Tr[XD(\frac{I}{2})] \times (2\%/72\%)$, where the four terms can be obtained by running the noisy quantum device. By

computing each term on the noisy quantum device and substituting results into the formula, we can obtain the ideal expectation value.

For a computation with multiple gates, the state preparation, measurement, and each gate can be treated in a similar way. Then, the formula of the ideal expectation value, i.e., a weighted summation of noisy computations has exponential terms with respect to the gate number. Therefore, instead of evaluating each term, we compute the summation using the Monte-Carlo method.

In this example, we consider the depolarizing error model. The decomposition can be applied to general error models without correlations. The decomposition formula is obtained by inverting the noise. For the gate $G$, the noise is $N = 80\%[I] + 20\%D$, and $G = NG^{id}$. The inverse of the noise is $N^{-1} = ([I] - 20\%D)/80\%$. Then, the ideal gate $G^{id} = N^{-1}G = (G - 20\%D)/80\%$.

**Analysis on residual errors**. Theoretically, the error mitigation technique, combining probabilistic error cancellation and gate set tomography, is capable of completely rectifying the effect of errors in the estimation of expectation values. However, in our experiment, the effective error rates after error mitigation are $(1.44 \pm 5.28) \times 10^{-5}$ and $(0.96 \pm 0.10) \times 10^{-3}$ in the single-qubit and two-qubit cases, respectively. Generally speaking, the reasons for the residual errors include the Pauli error assumption, time-correlated systematic drift, and crosstalk errors between qubits.

In the single-qubit case, the residual errors mainly come from the introduction of the Pauli error model. To quantify the non-Pauli error rate, we simulate the dynamics of the same random sequences as those used in the experiment with the characterized experimental PTMs, which are obtained under the Pauli error assumption. The experimental and simulated data of average fidelity are shown in Supplementary Fig. 2a, which are then numerically fitted to extract the error rates. The difference between the simulated and experimental error rates for single-qubit gates is $1.41 \times 10^{-5}$, which are of the same order of the residual error rate in the single-qubit case. Meanwhile, the data show that the time-correlated systematic drift has negligible effect and cannot be faithfully quantified within experimental and fitting errors.

In our experiment, we implement two different two-qubit gates, i.e., $MS_{YY}$ and $MS_{ZZ}$ gates. To quantify the residual errors from the Pauli error assumption, we compare the dynamics of the simulated and experimental random two-qubit sequence, where the simulation is based on the characterized PTMs with the Pauli error assumption. The difference between the simulated and experimental error rates gives the estimation of the non-Pauli residual error rate, which is about $0.20 \times 10^{-3}$. As to the crosstalk errors, the situations for $MS_{YY}$ and $MS_{ZZ}$ gates are quite different because of different implementation schemes. Specifically, a $MS_{ZZ}$ gate is implemented by a $MS_{YY}$ gate sandwiched by proper single-qubit gates, which introduce qubit-crosstalk errors. We model the crosstalk effect by measuring an effective Rabi frequency $\Omega_{eff}$ on the neighboring ion induced by leakage laser intensities when a single-qubit gate is being implemented by lasers focused on one of the ions. The ratio $\Omega_{eff}/\Omega$, with $\Omega$ being the Rabi frequency of the target ion, thus quantifies the severity of crosstalk errors. As shown in Supplementary Fig. 4a, we numerically simulate the state fidelities of the original and error-mitigated $MS_{YY}$ and $MS_{ZZ}$ gates. As expected, the numerical results show that $MS_{YY}$ gates, either original or error-mitigated ones, are insensitive to the crosstalk errors, whereas the fidelities of $MS_{ZZ}$ gates degrade as the severity of crosstalk errors increases. According to the numerical results, the crosstalk residual error rate is about $0.68 \times 10^{-3}$ at the experimental level of qubit crosstalk. Finally, the remaining part of the residual error rate, $0.08 \times 10^{-3}$, is attributed to the time-correlated systematic drift.

## Data availability

The data that support the findings of this study are available from the corresponding author upon reasonable request.

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

## Acknowledgements

This work was supported by the National Key Research and Development Program of China under Grant numbers 2016YFA0301900 and 2016YFA0301901, and the National Natural Science Foundation of China Grant numbers 11574002, 11504197, 11875050, and 11974200. Y.L. also acknowledges the support by NSAF Grant number U1930403.

## Author contributions

S.Z., Y. Lu, K.Z. and W.C. developed the experimental system. Y. Li proposed theoretical frame of the work. S.Z. and J.-N.Z. investigated the actual schemes for the experimental realization. S.Z. led the date taking. K.K. supervised the project. S.Z., Y. Li, J.-N.Z. and K.K. wrote the manuscript.

## Competing interests

The authors declare no competing interests.
