## [Peer Review File · Nature Communications]

Reviewers' Comments:

Reviewer #1:

Remarks to the Author:

The manuscript describes an experimental implementation of the recently proposed error mitigation technique, for a single qubit and two qubit gates. To implement this technique authors characterize the gate using a tomographic technique and then cancel these errors by randomly adding quantum gates to the sequence of gates. Authors claim that this technique increases effective gate fidelities by an order of magnitude, compared to the uncorrected gate.

While the results presented here are very interesting, I find the presentation of what is actually done too technical and confusing. In the introduction the authors compare their error mitigation technique with the error correction algorithms. I think this analogy is somewhat misleading: the error correction algorithms help to preserve quantum state in the presence of noise. The methods presented by authors (as far as I understand the manuscript) help to more precisely estimate expectation value of some observable based on the measured expectation values for original quantum circuit and some set of properly modified quantum circuits. These circuits and the recipe to calculate the final expectation value is chosen based on characterization of the imperfections of physical quantum gates.

The error mitigation technique does not actually keep the state error free or protect it from decoherence, it is an algorithm designed to cancel the errors introduced by the imperfection of the quantum gates when the expectation value of some observable are measured. Unfortunately it is not very easy to understand it after the first reading of the manuscript, and various statements like "We observe dramatic improvements on effective fidelities of single and two-qubit gates" and "The noise part of an experimental quantum operations can be formally reversed by random experimental basis operations ... " do not make it easier.

I think introduction and explanation of what was actually done should be significantly modified to improve clarity and avoid overselling the results.

In addition, I have a few more technical remarks.

1. Figure 4. Colors of the curves and shapes of data points on the plots are not defined. In addition, several data points on the Figure 4a are clearly above 1. By definition the fidelity of the quantum state can not exceed one. Since this figure is the main result of the manuscript, it deserves much more detailed explanations.

2. Methods section refers to the Figures S1, S2, etc that do not exist in the manuscript.

In conclusion, I believe the technique demonstrated in the manuscript is very useful to improve precision of the near term quantum computers, and the manuscript will be interesting for the readers of the Nature Communications. I recommend publication of the manuscript, if the clarity of the presentation is significantly improved and remarks above are addressed.

Reviewer #2:

Remarks to the Author:

Quantum computing experiments suffer from inaccuracies and decoherence. In principle logical error rates can be suppressed using quantum error correcting codes, but error rates at present are above the threshold necessary to apply these techniques. New methods, collectively called error mitigation, enable more accurate estimation of observables in the presence of noise. Although these methods do not scale, they have been shown to improve results in practice, provided that the computation occurs within the coherence time and other assumptions of each method are satisfied.

In this paper, the authors experimentally demonstrate (using trapped ions) an error mitigation technique called probabilistic error cancellation. This technique improves estimates of observables by using a quasi probability method to simulate a nonphysical inverse of each noise channel. The technique is applied to computations consisting of random sequences of gates, and it is shown that the estimated effective gate fidelity of one- and two-qubit gates significantly improves. Although error mitigation has been experimentally demonstrated, as far as I am aware this is one of the first published experiments to apply this particular error mitigation technique.

I feel that the authors oversell error mitigation techniques by not clearly stating their domain of applicability. As written, the paper gives the impression that we shouldn't bother with quantum error correction because we can "surpass the break-even point" with error mitigation with "no additional qubit resources". This potentially leaves the wrong impression. The authors should mention some explicit caveats and costs near the corresponding claims.

The technical details in the paper are not sufficiently clear to consider accepting the paper without first suggesting revision. Please see the specific comments below.

Page 3, figure 2: There is only one process matrix per gate. Are there significant differences in Gram or process matrices across qubits (eg depending on their position in the ion chain)? Is there any cross-talk when applying single-qubit gates in parallel/series? Since this is only a two-qubit experiment, perhaps these effects are not significant. If so please comment.

Page 3, figure 2: The meaning of the left and right columns are not explained. My guess is that they show the PTM reconstruction and the difference with the ideal PTM, respectively, but this should be stated somewhere.

Page 3, paragraph 2: The states $|1\rangle_X$ and $|1\rangle_Y$ in the set S_n are not defined. My guess is they are eigenstates of Pauli operators but again it should be stated.

Page 3, paragraph 3 ("For single-qubit randomized benchmarking ..."): This is very confusingly written. It seems like there are 3-4 ideas introduced at once: randomized benchmarking, gate-set tomography, how to model errors as Paulis operators, and how to estimate the parameters of the Pauli channels. This also seems like a crucial paragraph since it connects standard ideas with the authors' specific approaches. For example, is maximum likelihood estimation used both to reconstruct the PTM and to estimate the Pauli channel parameters? Can the authors include a reference to the estimation procedure(s) to clarify what is actually done?

Page 4, paragraph 3: It would help convince me that the procedures are correct if the authors could include (say in the methods section) a brief review of the error cancellation technique and the details of how it is applied.

Page 4, paragraph 3: Could the authors expand on how SPAM errors are removed in the two-qubit case?

Page 5, paragraph 1: What procedure is used for randomized benchmarking? It is not explicitly given in the paper as far as I can tell and does not seem to be standard. What does it mean "We post-select the uniformly-generated random sequences to minimize the projection error ..."?

Page 4, paragraph 1: It would be valuable to review briefly how you estimate fidelities since their values are key to your conclusion. Also, what does it mean on page 5 that you “end up with 4 random sequences for each sequence length L ”?

Figure 4: What is the curvature in the randomized benchmarking data for sequences of length up to 30? It does not seem to fit to an exponential. Why do you choose gate sequences of up to length 64 and 6 respectively?

First paragraph page 5: How are the signs of the quasiprobabilities integrated into the measurement results?

Last paragraph page 6: It would be helpful to refer to the discussion in the Methods section to justify the error budget discussion.

Reviewer #3:

Remarks to the Author:

The manuscript by Zhang et al reports the demonstration of error mitigation resulting in an effective reduction of the average error rate for sequences of single and two qubit gate circuits and is applicable in expectation estimation tasks. The method works by carrying out gate set tomography in order to characterise relevant errors and then use this information in the implementation of a gate sequence. By interleaving correction operations, which are based on gate set tomography, the authors accomplish to significantly improve the average fidelity of single and two qubit gate sequences. This method could be applied to VQE and as such might prove very useful. I think this is very interesting work and suitable for publication in Nature Communications. The manuscript is well written and the methods section provides sufficient information concerning the errors and residual errors after implementation of error mitigation. As such, I recommend publication in Nature Communications.

Reviewer #1 (Remarks to the Author):

Comments

The manuscript describes an experimental implementation of the recently proposed error mitigation technique, for a single qubit and two qubit gates. To implement this technique authors characterize the gate using a tomographic technique and then cancel these errors by randomly adding quantum gates to the sequence of gates. Authors claim that this technique increases effective gate fidelities by an order of magnitude, compared to the uncorrected gate.

While the results presented here are very interesting, I find the presentation of what is actually done too technical and confusing. In the introduction the authors compare their error mitigation technique with the error correction algorithms. I think this analogy is somewhat misleading: the error corrections algorithms help to preserve quantum state in the presence of noise. The methods presented by authors (as far as I understand the manuscript) helps to more precisely estimate expectation value of some observable based on the measured expectation values for original quantum circuit and some set of properly modified quantum circuits. These circuits and the recipe to calculate the final expectation value is chosen based on characterization of the imperfections of physical quantum gates.

The error mitigation technique does not actually keep the state error free or protect it from decoherence, it is an algorithm designed to cancel the errors introduced by the imperfection of the quantum gates when the expectation value of some observable are measured. Unfortunately it is not very easy to understand it after the first reading of the manuscript, and various statements like “We observe dramatic improvements on effective fidelities of single and two-qubit gates” and “The noise part of an experimental quantum operations can be formally reversed by random experimental basis operations ... “ do not make it easier.

I think introduction and explanation of what was actually done should be significantly modified to improve clarity and avoid overselling the results.

Our reply

We appreciate your comment of *“the results presented here are very interesting”*. Though the referee said that our presentation is confusing, he/she clearly and correctly understands the value of our works as commented *“The error mitigation technique does not actually keep the state error free or protect it from decoherence, it is an algorithm designed to cancel the errors introduced by the imperfection of the quantum gates when the expectation value of some observable are measured.”*

In this revision, we changed the first and second paragraphs of the introduction to avoid any confusion or misleading about the analogy of our error mitigation to error corrections algorithms. We only compare them in the relation to estimation of expectation values not for any other context.

Besides, we include more explanation and change the sentences mentioned by the referee as follows to avoid any potential ambiguities: “We observe dramatic improvements on effective gate fidelities of single- and two-qubit gates by an order of magnitude to those of physical gates. Here, the effective gate fidelities are obtained by fitting the corresponding expectation values estimated with error mitigation, which are not actual physical gate fidelities.” In the caption of Fig 1, the sentence “The noise part of an experimental quantum operations can be formally reversed by random experimental basis operations ... “ is changed to “The PTM of an ideal quantum gate can be expanded as a quasi-probability distribution over random gate sequences consisting of the experimental gate and one of the experimental basis operations, Pauli operations in the experiment. The error-mitigated estimation of the expectation value is then obtained by the Monte-Carlo method (see Methods note 4).”

Comments

In addition, I have a few more technical remarks.

1. Figure 4. Colors of the curves and shapes of data points on the plots are not defined. In addition, several data points on the Figure 4a are clearly above 1. By definition the fidelity of the quantum state cannot exceed one. Since this figure is the main result of the manuscript, it deserves much more detailed explanations.

Our reply

We include definitions for “*colors of the curves and shapes of data points*” in the caption.

The main reasons of some fidelity-data-points exceeding one after error mitigation come from the facts that 1) no principle of error-mitigation algorithm forces the fidelity of the state below 1, since the rescaling factor C is larger than 1 and 2) the number of sampling is not large enough. Actually even in an ideal situation that the number of sampling is infinite, the average value of fidelity would be below 1, but some part of fidelity distribution can be larger than 1. For the purpose of evaluating the performance of the error mitigation, we believe our numbers of samplings are good enough.

To explain that several data points above 1, we add the following sentence in the caption. “Some of the fidelities with error mitigation are larger than 1 because of the rescaling factor $C > 1$ (see Main Text and Methods note 4) and the limited sampling for data points. Although the current protocol does not guarantee a physical outcome, the error mitigation procedure shifts the distribution of the computation result towards the true value with large enough sampling.”

2. Methods section refers to the Figures S1, S2, etc that do not exist in the manuscript.

Our reply

We change the figure labels to make them consistent with the manuscript.

Comments

In conclusion, I believe the technique demonstrated in the manuscript is very useful to improve precision of the near term quantum computers, and the manuscript will be interesting for the readers of the Nature Communications. I recommend publication of the manuscript, if the clarity of the presentation is significantly improved and remarks above are addressed.

Our reply

We appreciate the referee acknowledges the importance of work and recommend the publication to Nature Communications. We also thank the referee for the valuable comments to improve our presentation of the works. We believe this revised manuscript clearly addresses all the points of the referee and much more clearly presents the results.

Reviewer #2 (Remarks to the Author):

Comments

Quantum computing experiments suffer from inaccuracies and decoherence. In principle logical error rates can be suppressed using quantum error correcting codes, but error rates at present are above the threshold necessary to apply these techniques. New methods, collectively called error mitigation, enable more accurate estimation of observables in the presence of noise. Although these methods do not scale, they have been shown to improve results in practice, provided that the computation occurs within the coherence time and other assumptions of each method are satisfied.

In this paper, the authors experimentally demonstrate (using trapped ions) an error mitigation technique called probabilistic error cancellation. This technique improves estimates of observables by using a quasi probability method to simulate a nonphysical inverse of each noise channel. The technique is applied to computations consisting of random sequences of gates, and it is shown that the estimated effective gate fidelity of one- and two-qubit gates significantly improves. Although error mitigation has been experimentally demonstrated, as far as I am aware this is one of the first published experiments to apply this particular error mitigation technique.

Our reply

We appreciate your pertinent understanding.

Comments

I feel that the authors oversell error mitigation techniques by not clearly stating their domain of applicability. As written, the paper gives the impression that we shouldn't bother with quantum error correction because we can “surpass the break-even point” with error mitigation with “no additional qubit resources”. This potentially leaves the wrong impression. The authors should mention some explicit caveats and costs near the corresponding claims.

Our reply

In the revision, we more clearly state the domain of our error mitigation applicability in abstract, introduction and discussion. In the abstract, we remove the “surpass the break-even point” to avoid unnecessary confusion, and include “on the estimation of expectation values” in the sentence of “without additional qubit resources”. In the first and second paragraphs of the introduction, we repeated several times the application of the error mitigation is limited to the estimation of the expectation values. Moreover, in the discussion, we add several sentences as explicit caveats. We believe these modifications clearly address the referee's concern of overselling of our results.

Comments

Page 3, figure 2: There is only one process matrix per gate. Are there significant differences in Gram or process matrices across qubits (eg depending on their position in the ion chain)?

Is there any cross-talk when applying single-qubit gates in parallel/series? Since this is only a two-qubit experiment, perhaps these effects are not significant. If so please comment.

Our reply

We clarify that the single-qubit experiments shown in Fig. 2(a)(b) are performed in a single-qubit system, and therefore, there is only one Gram matrix and one Pauli transfer matrix for each gate. In the two-qubit system, the 4x4 Pauli transfer matrices (PTMs) for single-qubit gates on both qubits are not significantly different from those obtained in the single-qubit system. It is mainly because the 4x4 PTMs cannot characterize the influences on the other qubit.

However, we do expect the existence of cross-talk errors when applying single-qubit gates in a two-qubit system, which is the main limitation to further improvement of the accuracy of the estimation of the expectation values. We estimate the cross-talk errors by measuring the leakage laser intensity on the other ion, which is 1.6 (+0.1)%. Then, we simulate the state fidelity with the effect of laser leakage, and estimate the crosstalk error of 0.68×10^{-3} with MSzz case, as presented in Supplementary Fig. S4a. Since the circuit of MSzz gate contains 4 single-qubit operations (we treat them as being applied in series) as shown in Fig. S4b, we believe it provides the upper bound of the crosstalk errors. We note that we do not have Gram matrix in the two-qubit case, since we introduce the calibration technique proposed in Ref (31) to remove the relatively large SPAM errors in consideration of efficiency. We add such necessary explanation in the caption of figure 2.

Comments

Page 3, figure 2: The meaning of the left and right columns are not explained. My guess is that they show the PTM reconstruction and the difference with the ideal PTM, respectively, but this should be stated somewhere.

Our reply

We include the meanings of the left and right columns in the caption of Fig. 2 as “In each subfigure, the left column shows the experimentally-obtained matrices and the right column shows the difference between the experimental and the ideal matrices, i.e. $R_G - R_G^{\text{id}}$ with G being one of the quantum operations being characterized.”. We also add labels in Fig. 2 to make it clearer.

Comments

Page 3, paragraph 2: The states $|1\rangle_X$ and $|1\rangle_Y$ in the set S_n are not defined. My guess is they are eigenstates of Pauli operators but again it should be stated.

Our reply

We add the definitions of $|1\rangle_X$ and $|1\rangle_Y$.

Comments

Page 3, paragraph 3 ('For single-qubit randomized benchmarking ...'): This is very confusingly written. It seems like there are 3-4 ideas introduced at once: randomized benchmarking, gate-set tomography, how to model errors as Paulis operators, and how to estimate the parameters of the Pauli channels. This also seems like a crucial paragraph since it connects standard ideas with the authors' specific approaches. For example, is maximum likelihood estimation used both to reconstruct the PTM and to estimate the Pauli channel parameters? Can the authors include a reference to the estimation procedure(s) to clarify what is actually done?

Our reply

This paragraph is mainly about how we perform the gate set tomography and obtain the Pauli transfer matrices of the gates in the gate set by the maximum likelihood estimation. The randomized benchmarking is not the main point of the paragraph. We briefly mention the randomized benchmarking here only to justify the constituents of the gate set, and the concrete procedure of randomized benchmarking and how to apply the probabilistic error cancellation technique are described in paragraph 10 starting with the sentence of "We benchmark the performance ..."

The gate set tomography itself is a general tool for quantum system characterization, which does not impose any assumption on the system and the quantum process to be characterized. In our experiment, however, we assume there are only Pauli errors, which is a good assumption in our trapped-ion system. Based on this Pauli-error assumption, we make an ansatz for the Pauli transfer matrix of each gate, with Pauli error rates being variational parameters. For each experimental setting $\langle\langle E_i | R_{\{G_j\}} | \rho_k \rangle\rangle$, we calculate the ansatz prediction of the average $m_{\{ijk\}}$, which is a function of the Pauli error rates. Then we repeatedly implement each of the experimental settings in the trapped-ion system and experimentally obtain the average $\bar{m}_{\{ijk\}}$ and the standard deviation $\Delta_{\{ijk\}}$. The likelihood function, which again is a function of the Pauli error rates of all the gates in the gate set, is defined such that it is maximized when the ansatz prediction $m_{\{ijk\}}$ and the observed $\bar{m}_{\{ijk\}}$ are equal. Thus, we simultaneously obtain the Pauli error rates for all of the gates in the gate set by maximizing the likelihood function, i.e. the maximum likelihood estimation. Finally, the Pauli transfer matrix of each gate is constructed by plugging the Pauli error rates into the ansatz. Actually, the Ref. [25] contains a comprehensive study on the gate set tomography, which we have cited many times in the manuscript.

To improve the readability of the paragraph, we briefly summarize the method in the main text and we move those detailed explanations above to Methods to comprehensively describe our maximum likelihood gate-set tomography with Pauli error assumption.

Comments

Page 4, paragraph 3: It would help convince me that the procedures are correct if the authors could include (say in the methods section) a brief review of the error cancellation technique and the details of how it is applied.

Our reply

We add Methods note 4 and note 5 to provide a brief review and details of the error cancellation technique as suggested for the self-consistency, though it is already discussed in detail in the Ref. [18].

Comments

Page 4, paragraph 3: Could the authors expand on how SPAM errors are removed in the two-qubit case?

Our reply

Ref. [31], which proposes the technique, gives a clear representation of the general procedure of removing arbitrary uncorrelated SPAM errors. In our experiment, we introduce this technique in the two-qubit case. We prepare the system in the states $|00\rangle$ and $|11\rangle$, and measure the state fidelities of $|0\rangle$ and $|1\rangle$ for both qubits. The infidelities of these states give the SPAM error probability associated with each measurement outcome, which can then be used to remove the SPAM errors by data processing.

We add necessary explanation at the end of this paragraph.

Comments

Page 5, paragraph 1: What procedure is used for randomized benchmarking? It is not explicitly given in the paper as far as I can tell and does not seem to be standard. What does it mean “We post-select the uniformly-generated random sequences to minimize the projection error ...”?

Our reply

The single-qubit randomized benchmarking adopted in our experiment is standard, of which the procedure is same as that described in Ref. [22].

In the two-qubit case, we have described the random sequences by the sentence “Each sequence contains L two-qubit gates uniformly drawn from the set $\{MS_{YY}, MS_{ZZ}\}$ with interleaving single-qubit gates.” The distribution of random sequences is different from the one used in the standard randomized benchmarking. Elementary gates in our sequences are not uniformly drawn from the entire two-qubit Clifford group. We observe the improvement by the error mitigation for each gate sequence. Therefore, the difference in the distribution is not likely to change the conclusion.

The sentence “We post-select the uniformly-generated random sequences to minimize the projection error...” means we choose the random sequences whose ideal final states are eigenstates of $Z^{\otimes 2}$. We change the sentence accordingly to make it clear.

Comments

Page 4, paragraph 1: It would be valuable to review briefly how you estimate fidelities since their values are key to your conclusion. Also, what does it mean on page 5 that you “end up with 4 random sequences for each sequence length L ”?

Our reply

The fidelities we have shown in Fig. 4 are average state fidelities. For both of the single-qubit and the two-qubit cases, we first generate 4 random gate-sequences for each sequence length. Then we implement each of the sequences and measure the state fidelity between the ideal and experimental final states, with which we calculate the average state fidelities by averaging over the random sequences of the same length.

The sentence “end up with 4 random sequences for each sequence length L ” means “choose 4 random sequences ...”. We modify the sentence as follows to avoid confusion as follows.

“For the two-qubit case, we select four gate sequences as benchmarking computations for each length L .”

The sequence is selected under the restriction that the ideal final state is an eigenstate of $Z^{\otimes 2}$.”

Comments

Figure 4: What is the curvature in the randomized benchmarking data for sequences of length up to 30? It does not seem to fit to an exponential. Why do you choose gate sequences of up to length 64 and 6 respectively?

Our reply

Considering the fact that we only average over 4 different random sequences, we believe the curvature is due to statistical fluctuation or time-correlated systematic drift in the system. However, in our experiment, we apply error mitigation for each of the sequences separately and we clearly see the improvement of effective gate fidelities.

We choose gate sequences of up to length 64 for the single-qubit case and 6 for the two-qubit case, because the experimental data are adequate both to extract the physical gate fidelity and to show the improvement of applying the probabilistic error cancellation technique.

Comments

First paragraph page 5: How are the signs of the quasiprobabilities integrated into the measurement results?

Our reply

The sign of the quasiprobabilities is multiplied with the measurement outcome when calculating the average. We add Methods note 4 to provide a detailed procedure.

Comments

Last paragraph page 6: It would be helpful to refer to the discussion in the Methods section to justify the error budget discussion.

Our reply

We refer to Methods note 6, which discusses the error budget, in the last paragraph as suggested.

Reviewer #3 (Remarks to the Author):

Comments

The manuscript by Zhang et al reports the demonstration of error mitigation resulting in an effective reduction of the average error rate for sequences of single and two qubit gate circuits and is applicable in expectation estimation tasks. The method works by carrying out gate set tomography in order to characterise relevant errors and then use this information in the implementation of a gate sequence. By interleaving correction operations, which are based on gate set tomography, the authors accomplish to significantly improve the average fidelity of single and two qubit gate sequences. This method could be applied to VQE and as such might prove very useful. I think this is very interesting work and suitable for publication in Nature Communications. The manuscript is well written and the methods section provides sufficient information concerning the errors and residual errors after implementation of error mitigation. As such, I recommend publication in Nature Communications.

Our reply

We greatly appreciate the referee for the accurate understanding of our work and the recommendation of the publication.

Reviewers' Comments:

Reviewer #1:

Remarks to the Author:

In the revised version of the manuscript authors have addressed all my comments and suggestions. As far as I am concerned, the manuscript can be published in Nature Communications.

Reviewer #2:

Remarks to the Author:

The authors have addressed all of the comments. I recommend publishing the manuscript.